# Causal Effects of Prenatal Exposure to PM_2.5_ on Child Development and the Role of Unobserved Confounding

**DOI:** 10.3390/ijerph16224381

**Published:** 2019-11-09

**Authors:** Viola Tozzi, Aitana Lertxundi, Jesus M. Ibarluzea, Michela Baccini

**Affiliations:** 1Department of Statistics, Computer Science, Applications, University of Florence, 59 50134 Florence, Viale Morgagni, Italy, viola.t@hotmail.it; 2BIODONOSTIA Health Research Institute, 20014 San Sebastian, Spain; aitana.lertxundi@ehu.eus (A.L.); mambien3-san@euskadi.eus (J.M.I.); 3Biomedical Research Centre Network for Epidemiology and Public Health (CIBERESP), 28029 Madrid, Spain; 4Faculty of Medicine, University of the Basque Country (UPV/EHU), 48940 Leioa, Spain; 5Sub-Directorate for Public Health of Guipúzcoa, Department of Health, Government of the Basque Country, 20013 San Sebastian, Spain; 6Faculty of Psychology, University of the Basque Country (UPV/EHU), 20018 San Sebastian, Spain

**Keywords:** child development, airborne particles, propensity score matching, sensitivity analysis, bias analysis, Monte Carlo simulations

## Abstract

Prenatal exposure to airborne particles is a potential risk factor for infant neuropsychological development. This issue is usually explored by regression analysis under the implicit assumption that all relevant confounders are accounted for. Our aim is to estimate the causal effect of prenatal exposure to high concentrations of airborne particles with a diameter < 2.5 µm (PM2.5) on children’s psychomotor and mental scores in a birth cohort from Gipuzkoa (Spain), and investigate the robustness of the results to possible unobserved confounding. We adopted the propensity score matching approach and performed sensitivity analyses comparing the actual effect estimates with those obtained after adjusting for unobserved confounders simulated to have different strengths. On average, mental and psychomotor scores decreased of −2.47 (90% CI: −7.22; 2.28) and −3.18 (90% CI: −7.61; 1.25) points when the prenatal exposure was ≥17 μg/m^3^ (median). These estimates were robust to the presence of unmeasured confounders having strength similar to that of the observed ones. The plausibility of having omitted a confounder strong enough to drive the estimates to zero was poor. The sensitivity analyses conferred solidity to our findings, despite the large sampling variability. This kind of sensitivity analysis should be routinely implemented in observational studies, especially in exploring new relationships.

## 1. Introduction

Many studies around the world documented long term and short term effects of air pollution on population mortality and morbidity [1,2]. Recently, several researches focused on the effects of prenatal exposures to air pollution on children’s neuropsychological development [3]. Harris et al. [4] analyzed 1109 mother–child pairs from Eastern Massachusetts (USA) concluding that parental proximity to major roadways may negatively influence performance across a range of mental domains in childhood. Guxens et al. [5] found significant associations between high levels of nitrogen dioxide (NO_2_) and benzene and infant neurodevelopment in the Spanish cohorts participating in the INMA (Infancia y Medio Ambiente, the Spanish for Childhood and Environment) Project. High levels of NO_2_ were found to be associated also with delayed psychomotor development in children between one and six years of age [6]. Lertxundi et al. [7,8] investigated the effect of prenatal exposure to airborne particles with a diameter < 2.5 µm (PM_2.5_) and NO_2_ on infant cognitive and psychomotor development and on the cognitive functions in children of 4–6 years of age, reporting the existence of negative associations.

All these studies accounted for the confounding effect of several mother and child characteristics through the specification of regression models, under the implicit assumption that all relevant confounders were known and observed (assumption of unconfoundedness [9]). However, given the complex nature of the problem, the presence of residual confounding due to unknown or unmeasured factors related to the familiar and environmental context where children live should not be excluded. In the literature, the problem related to the presence of unobserved confounders and the evaluation of the associated bias has been addressed through sensitivity analyses that check whether and to what extent the estimated effect or association is robust to possible deviations from unconfoundedness [10,11,12,13,14,15,16,17,18]. Usually, the aim is to assess how strong an unobserved confounder would have to be to substantially change the study results. However, despite the importance of this issue, quantitative evaluations of the potential bias associated with residual confounding have rarely received consideration in epidemiological applications (see chapter 19 in [10]). In the present paper, with reference to a birth cohort of the province of Gipuzkoa [7], we estimated the causal effect of high PM_2.5_ exposures during maternal pregnancy on child mental and psychomotor outcomes at 15 months of age, within a potential outcomes approach to causal inference [19]. Then, we checked the robustness of the results to the possible presence of unobserved confounders through a sensitivity analysis based on simulations. In particular, we compared the actual causal effect estimates obtained by using propensity score (PS) matching [9], to those obtained after controlling also for simulated unobserved confounders of different strengths, according to the method proposed by Ichino et al. [11] and implemented in the *sensatt* function of STATA (StataCorp LP., College Station, TX, USA) [20,21]. 

## 2. Materials and Methods 

### 2.1. Data

As part of the Environment and Childhood Research Network (INMA network) [22], the birth cohort of Gipuzkoa was recruited between May 2006 and January 2008 among the residents of a study area of 519 km^2^ spanning three narrow valleys: high Urola-Goierri, mid Urola and high Oria. The total population of the area was approximately 88,000 inhabitants, spread across 25 small localities. Data on exposures, outcomes and potential confounders were available for 438 mother–child pairs. Children’s mental and psychomotor developments were assessed at around 15 months of age (range 13–18 months), according to the Bayley Scales of Infant Development [23], by one of two trained neuropsychologists, blinded to the child’s exposure status. The raw scores were standardized to a mean of 100 points with a standard deviation of 15 points, with higher scores indicating better developments. Data on the mother’s characteristics and habits were obtained through questionnaires administered during the first and third trimesters of pregnancy. The daily levels of PM_2.5_ in the area were measured from the beginning of the study until the last birth by three fixed-site samplers belonging to the Air Quality Network of the Basque Government and three Digitel DHA-80 high-volume aerosol samplers, two of which were monthly rotated in different locations (Figure 1). 

Due to the presence of moving monitors, daily measurements were characterized by the presence of design-missing values. With the aim to assess the average exposure of the mother–child couples during pregnancy, we used a multiple imputation (MI) procedure starting from the weekly data. First, we derived weekly means from the daily levels of PM_2.5_ measured in each location. Considering a weekly mean as missing when daily measurements were missing for more than two days during the week, the percentage of missing weekly averages during the study period was around 30%. Then, under the Missing At Random (MAR) assumption, we applied an MI procedure [24] on the incomplete data set including weekly averages concentrations of the air pollutants measured by the monitors, meteorological variables and seasonality indicators. In this way, we generated five imputed data sets, thus five imputed values for each missing weekly mean of PM_2.5_. Finally, we averaged over weekly means, obtaining five imputed values of the average level of PM_2.5_ during the pregnancy for each mother–child couple. We assumed that the mother–child couples were exposed to PM_2.5_ levels measured at the monitoring site located in their town or in the closest one. For more detail on the MI procedure see [7]. 

The study was approved by the hospital ethics committee and all participating mothers provided informed consent.

### 2.2. Notation

Unless otherwise specified, the notation and the methods described hereinafter refer to a single imputed data set.

Being *i =* 1, *N* the indicator of the child, let *E_i_* be a binary exposure indicator, which was equal to 0 if the average level of PM_2.5_ during the pregnancy was <17 µg/m^3^ (control) and equal to 1 if it was ≥17 µg/m^3^ (treatment). The threshold of 17 µg/m^3^, which approximated the median exposure of the units across the imputed data sets, was fixed a priori in order to avoid post-hoc data dredation [25]. Let *Y_i_* be the observed mental or psychomotor score for child *i* and **X***_i_* a vector of observed covariates.

According to the potential outcome approach to causal inference, under the Stable Unit Treatment Value Assumption, we associated with each child two potential outcomes *Y_i_*(1) and *Y_i_*(0), representing the score under treatment and under control [9]. For each child it was possible to observe only one of the two potential outcomes: for children belonging to the treated group we observed only *Y*(1) (*Y_i_*(1) = *Y_i_* if *E_i_* = 1); for children belonging to the control group we observed only *Y*(0) (*Y_i_*(0) = *Y_i_* if *E_i_* = 0).

### 2.3. Research Question and Estimands

The research question we were interested in was the following: what would have been the mental and psychomotor development of the children with prenatal exposure ≥17 µg/m^3^, if their exposure were been <17 µg/m^3^? In order to reply to this questions, we focused, separately for psychomotor and mental score, on the average causal effect of treatment on the treated (ATT), which compares the average score observed among the “treated” children, with the average score that we would have observed on them if they were not “treated”:(1)$$ATT=\frac{\sum_{i=1}^{N}Y_{i}\left(1\right)E_{i}}{\sum_{i=1}^{N}E_{i}}-\frac{\sum_{i=1}^{N}Y_{i}\left(0\right)E_{i}}{\sum_{i=1}^{N}E_{i}}=\frac{\sum_{i=1}^{N}Y_{i}E_{i}}{\sum_{i=1}^{N}E_{i}}-\frac{\sum_{i=1}^{N}Y_{i}\left(0\right)E_{i}}{\sum_{i=1}^{N}E_{i}}.$$

### 2.4. Assumption and ATT estimation

In order to estimate the ATT, we imputed the missing potential outcome *Y_i_*(0) in (1) through a matching procedure based on the PS [9]. This required two assumptions [26]:

(i) Unconfoundedness for controls: *E_i_*⊥*Y_i_*(0)|**X***_i_*. This assumption states that treatment assignment is independent of the potential outcome under control, conditional on the observed covariates. In other words, this assumption requires that there are no unobserved confounders.

(ii) Weak positivity assumption: P(*E_i_* = 1|**X***_i_* = **x**) < 1 for each *i*, for each **x**. This assumption implies that for each treated unit there is a similar untreated unit that can be used as a matched control.

Being the PS a balancing score, the conditions (i) and (ii) are valid also conditionally to the PS [26].

Under (i) and (ii), we applied a PS matching procedure for AT estimation which consisted of two steps:

(1) For each child, we estimated the PS as predicted value from a logistic model defined on the exposure *E_i_*, given **X***_i_* [26]. The model accounted for the following variables: child’s sex, year of birth, nursery attendance (yes, no), main caregiver (only mother, both parents, other relative, other caregiver), maternal education (secondary school or less, university), maternal work (non-manual worker, manual worker), mother’s age (<25, 25–34, 35+), mother’s body mass index (underweight, normal, overweight, obese), mother’s parity (0, 1+), mother’s vegetable and fruit intake during pregnancy (≤405 g/day, >405 g/day), smoking during pregnancy (yes, no), smoking at 32 weeks of pregnancy (yes, no), breastfeeding (yes, no), indicator of the neuropsychologist who evaluated the child (first, second evaluator). We included in the PS model not only pre-treatment variables, as usually recommended [27], but also some post-treatment variables (smoking and diet during pregnancy, child care mode, breastfeeding) which were not affected by the exposure and could be thought of as proxies for unmeasured pre-treatment covariates, mainly related to the socio-economic status of the family. We expect that adjusting for them could reduce the bias without inflating the variance [28,29].

Regarding the need of adjusting for the neuropsychologist indicator, it is worth noting that the probability of exposures ≥17 µg/m^3^ was very low (from 1% to 5%) for the children examined by the second neuropsychologist, who substituted the first one at the end of the study period. The observed decrease in the exposures over time was mainly due to the substantial reduction of the atmospheric level of PM_2.5_ in 2007, likely attributable to the high level of precipitations (58.8 mm in 2006 versus 126.5 mm in 2007) and to the closure of a polluting industry in the area in July of the same year (Appendix A) [30]. This, coupled with the fact that the scores assigned by the second psychologist were systematically higher than those assigned by the first one, made crucial to adjust for the neuropsychologist indicator.

Once the relevant covariates for the PS estimation have been defined, the key criterion for driving the specification of the PS model was to obtain PS estimates that balanced the covariate distribution between treated and controls [9]. According to the procedure implemented in the STATA command *pscore*, we checked the balancing property of the PS by comparing the covariate distribution between the two groups of exposure within strata of units with similar estimated PS [31]. We also checked for the weak positivity condition (ii), by the inspection of the PS distributions among treated and controls.

(2) On the basis of the estimated PS, we matched each treated child with the child in the control group having closest PS, $Y_{i}^{c}$. The same matched control could be used for different treated units, in order to reduce the bias due to inappropriate matching. Finally, the ATT was estimated as:(2)$$\hat{ATT}=\frac{\sum_{i=1}^{N}Y_{i}E_{i}}{\sum_{i=1}^{N}E_{i}}-\frac{\sum_{i=1}^{N}Y_{i}^{c}E_{i}}{\sum_{i=1}^{N}E_{i}}.$$

The standard error of $\hat{ATT}$ was estimated without accounting for the uncertainty around the estimated PS [31].

The estimation procedure was separately performed on each of the five multiple-imputed data sets, and the five results combined according to Rubin [32]. It is worth noticing that, for each multiple-imputed data set, the ATT was calculated on different treated populations. This required a cautionary interpretation of the ATT, which referred to different sets of children depending on the imputation.

### 2.5. Sensitivity Analysis

The Ichino’s sensitivity analysis, which is mainly employed in social science and economics, is based on Monte Carlo simulations [11]. Let us suppose that the unconfoundedness assumption (i) is not valid because of the presence of a binary unobserved confounder U. A value of U is attributed to each subject by sampling from a hypothetical distribution, then U is included in the set of variables used to estimate the ATT. This procedure can be repeated many times (e.g., 1000) and, at the end, an average ATT is estimated. Comparing this average ATT with the causal estimate obtained without adjusting for U, the robustness of the result to violation of the unconfoundedness assumption can be checked: if the ATT estimates before and after adjusting for U are close, the result can be considered robust to the presence of an unobserved confounder similar to U.

In the present paper, we explored the space of the possible unobserved confounders by generating different versions of U, which covered different hypotheses on confounding, as expressed by the associations of U with the outcome and with the exposure. According to Ichino et al. [11], indicating with $Y^{*}$ a binary version of the outcome (for example $Y^{*}=1$ if *Y* > 75th percentile, $Y^{*}=0$ otherwise), the association of U with the outcome (outcome effect) is measured by the odds ratio Γ comparing the odds of $Y^{*}=1$ when U is equal to 1 with the odds of $Y^{*}=1$ when U is equal to 0, among the controls: (3)$$\Gamma =\frac{\frac{P\left(Y^{*}\;=\;1|E\;=\;0,U\;=\;1,X\right)}{P\left(Y^{*}\;=\;0|E\;=\;0,U\;=\;1,X\right)}}{\frac{P\left(Y^{*}\;=\;1|E\;=\;0,U\;=\;0,X\right)}{P\left(Y^{*}\;=\;0|E\;=\;0,U\;=\;0,X\right)}}$$

The association of U with the exposure (selection effect) is measured by the odds ratio Λ comparing the odds of being exposed when U is equal to 1 with the odds of being exposed when U is equal to 0:(4)$$\;\Lambda =\frac{\frac{P\left(E\;=\;1|U\;=\;1,X\right)}{P\left(E\;=\;0|U\;=\;1,X\right)}}{\frac{P\left(E\;=\;1|U\;=\;0,X\right)}{P\left(E\;=\;0|U\;=\;0,X\right)}}$$

The *sensatt* function generates U starting from four tuning parameters: $p_{ij}=Pr\left(U=1|E=i,Y^{*}=j\right),\;i=0,1,\;j=0,1$, and calculates the odds ratios Γ and Λ from the simulations.

In our study, we focused on unobserved confounders with a distribution similar to the empirical distribution of relevant observed confounders, by deriving the *p_ij_* from the observed data. Successively, we explored the whole space of the hypothetical Us by setting $p_{ij}$ = 0.1, 0.25, 0.5, 0.75, 0.9, i=0,1, j=0,1, in order to search for the so-called “killer” confounder, i.e., the confounder U able to explain away the estimated effect, driving the estimated ATT to zero. In this second case, it is crucial to assess the plausibility of the configurations of *Γ* and *Λ* which lead to the killer confounder: if the killer configuration is unlikely, the ATT estimate can be considered robust to the violation of the unconfoundedness assumption.

All the sensitivity analyses were separately performed on each of the five multiple-imputed data sets and the results combined according to Rubin [32].

## 3. Results

The analyses were conducted on the subset of the 391 children without missing covariates. The covariates distribution in this subset did not substantially differ from that observed on the entire data set (Table 1).

The procedure for the balancing check implemented in the *pscore* function did not find a relevant difference between the covariate distribution of treated and controls within strata of children with similar estimated PS (results not reported). Moreover, the distributions of the estimated PS among treated and controls substantially overlapped (Appendix A), indicating that it was possible to find, for each treated, a matched control with similar PS, as stated by the positivity condition (ii). We would like to stress that the fact that the probability of being treated was close to zero for the children evaluated by the second neuropsychologist (Table 1) did not affect the assumption (ii).

The number of subjects in the high exposure group varied from 183 to 213 depending on the imputed data set. Similarly, the number of matched controls varied from 62 to 69. The population of the treated units, which the ATT referred to, was quite stable across imputed data sets: for more than 90% of the 391 analyzed children the exposure status did not change across the imputed data sets: 172 children were “always exposed” and 181 were “always unexposed” across the five imputed data sets (Appendix A). Likely, the exposure status was uncertain only for children with actual average prenatal exposure close to the threshold of 17 µg/m^3^. 

Combining the estimates of the ATT from the five imputations, we found that the expected scores under PM_2.5_ ≥17µg/m^3^ were lower than the average score under PM_2.5_<17 µg/m^3^, with final ATT estimates equal to −2.47 (90% CI: −7.22, 2.28) for the mental score and −3.18 (90% CI: −7.61, 1.25) for the psychomotor score. In all the analyses, the statistical variability was due to both the within and between-imputation variances, with the second being less than 10% of the first one (Appendix A). 

Table 2 and Table 3 report the results of the sensitivity analyses conducted by simulating several versions of U mimicking the behavior of the observed confounders. In doing this we selected some relevant categories of the confounders and simulated U according to the corresponding indicators (this was necessary because the *pscore* function permits only for the definition of binary unobserved confounders). The odds ratios *Γ* and *Λ* were calculated defining *Y** = 1 if *Y* > 75th percentile and 0 otherwise. Even if in principle the selection effect *Λ* should be the same in the mental and psychomotor scores analyses, the values reported in Table 1 and Table 2 are different because they arise from different simulations. The odds ratio *Γ* ranged from 0.20 to 2.25 for mental score and from 0.31 to 4.30 for psychomotor score; *Λ* ranged from 0.57 to very large values (>10^3^). This indicates that we were exploring a fairly wide class of possible unobserved confounders. 

Overall, including among the confounders in the PS matching, the simulated Us also did not bring to qualitatively different ATT estimates. Only when U mimicked the neuropsychologist indicator, generating low values of *Γ* and very large values of *Λ*, the estimated effects were fully explained away. It is worth noticing that extremely high values of *Λ* when U mimicked the neuropsychologist indicator originated from the fact that the probability of being treated for the children evaluated by the second neuropsychologist ranged, in the imputed data sets, from 1% to 5%.

The results obtained in searching for the killer confounder are reported in Figure 2. Unobserved confounders positively associated with the outcome and negatively associated with the exposure, or vice versa, could kill the effect if at least one of the odds ratios *Λ* and *Γ* was approximately lower than 1:10 or larger than 10:1. However, the plausibility of having omitted a confounder of such strength was poor in this context, especially considering that many relevant covariates had already been included in the PS model.

Selecting the 50th percentile of *Y* for the definition of *Y** did not lead to substantially different conclusions.

## 4. Discussion

Our results suggest that prenatal exposure to average levels of PM_2.5_ ≥ 17 µg/m^3^ had a negative effect on a child’s mental and psychomotor development at around 15 months of age, even if the confidence intervals of the effects estimates were wide, reflecting large sampling variability. 

Unobserved confounders moderately associated with both outcome and exposure could kill the effects in the sense of driving the point estimates of the causal effects to zero. However, the required associations were stronger than those calculated for most of the observed confounders, making the omission of such relevant factors quite implausible, in particular if the sensitivity analysis results are interpreted “in context”, i.e., accounting for number and relevance of the observed covariates which we already adjusted for [13]. In fact, the sensitivity analysis explores the robustness of the results to the existence of unmeasured confounders that act through pathways independent of the observed covariates. Therefore, the same strength of association required to an unobserved confounder to kill the effect can be indicative of large robustness if the original model accounts for many relevant covariates, as in the present paper, or of poor robustness if the original model accounts just for few confounding factors.

In interpreting the results of the sensitivity analysis, we focused on the point estimates of the effects and not on the confidence intervals limits as sometimes suggested (searching for possible unobserved confounders able to explain away the upper or lower limit of the interval, depending on the sign of effect estimate) [11,12]. This choice was motivated by our will to distinguish between the evaluation of bias and evaluation of uncertainty. In particular, we think that the sensitivity analysis should be performed regardless the wideness of the confidence interval and its position in respect to the null hypothesis, because its aim is to evaluate the possible presence of major bias due to omission of relevant confounders, which is different from evaluating the “significance” of the result. When the confidence interval of the estimate is wide as in our analysis, the robustness of the point estimate, if demonstrated, can confer a certain strength to the conclusions, despite the large sampling variability. As other kinds of selective reporting, limiting the sensitivity analysis to the subset of “significant” results may produce bias in the literature, with only studies with narrow confidence intervals evaluated for their robustness in respect to unobserved confounding. This issue is crucial especially in a perspective where, as desirable, this kind of procedure can become standard in observational studies. Finally, even if it refers to a specific case study, the sensitivity analysis may provide useful insights about the confounders to be accounted for or collected in future studies on the same research question, contributing to enhance the knowledge about the phenomenon of interest.

Ichino’s sensitivity analyses rely on the definition of a single binary unobserved confounder U, and on the assumption of no interaction between the effects of the exposure and U on the outcome. A sensitivity analysis that relaxes these assumptions has been recently proposed by Ding and VanderWeele [12,13], but it was not considered in this paper. A second drawback of the simulation method is its high computational demand. In fact, it requires that the statistical procedure used to estimate the effect is repeated *n* times in order to generate a distribution of the effect estimate adjusted for the hypothetical unobserved confounder. This is particularly expensive when complex analyses are required, as in our application, where an MI procedure is used. The computational burden of the simulation approach is particularly evident when the aim is to search for the killer confounder over the whole space of the hypothetical *U*s.

Our findings qualitatively agree with those obtained for the same cohort by Lertxundi et al. [7] using a regression approach. In Lertxundi et al. [7], the linearity assumption on the exposure-outcome relationship, together with the parametric specification of the exposure-confounders-outcome relationship, contributed to increase the precision of the effect estimates, but, at the same time, could have generated a certain amount of bias due to problems of inappropriate model specification. On the contrary, the PS matching protected us from possible bias due to nonlinearity or inappropriate modeling of the exposure-confounders-outcome relationship, but at the prize of a lower precision of the effect estimates. Anyway, the fact that different statistical methods brought to consistent results strengthens our conclusions regarding the negative effect of the exposure.

### Study Limitations

Our study has several limitations. First, we limited the analyses to the subset of children for which all covariates included in the PS model were observed. This affected the power of the analysis and could have brought a certain amount of bias, although it has been suggested that in several situations estimating the PS on the complete cases can yield a valid estimate of the causal effect in the subjects without missing values [33].

Second, exposure during pregnancy was not based on personal exposure, that, if available, could provide more accurate results. Always in relation with the exposure definition, because of to the small dimension of the area and the small number of available monitors, we preferred to associate to each mother–child couple the level of PM_2.5_ measured by the monitor located in the municipality of residence instead of using more complex exposure assessment procedure, e.g., kriging interpolation, to predict the level of the exposure at the residence address.

Finally, we treated continuous exposure as a binary one, after having defined an arbitrary threshold. Focusing on a binary version of the exposure allowed us to apply the sensitivity analysis as implemented in the *sensatt* function without complex adaptations, but at the same time led to a certain degree of arbitrariness related to the choice of the threshold for the exposure, and to loss of information which resulted in large sampling variability. A future promising extension of the analysis could use methods based on the generalized propensity score, which, treating the exposure as continuous, allows for estimating the whole exposure-response relationship [34,35]. 

## 5. Conclusions

Although the variability of the estimates was large, our findings indicated that prenatal exposure to high levels of PM_2.5_ had a negative effect on a child’s mental and psychomotor development in the analyzed cohort. The causal effect estimates appeared substantially robust to the possible presence of unmeasured confounders, even considering that the causal estimates were already adjusted for many relevant factors.

Checking the robustness of the estimates to the presence of unobserved confounders is very important in observational studies and should not be considered less relevant than quantifying the sampling uncertainty around the estimates. Especially when new relationships are investigated, sensitivity analyses aimed to evaluate the robustness of the results to the omission of relevant factors in the analysis should become a standard. In this paper, we proposed a simulation approach to be employed after PS matching that, even if computationally expensive, is implemented in standard statistical software and provides results easy to be interpreted. The use of other methods for sensitivity analysis, which eventually require less computational effort and rely on less stringent assumptions, should be explored as well.

## Figures and Tables

**Figure 1 ijerph-16-04381-f001:**
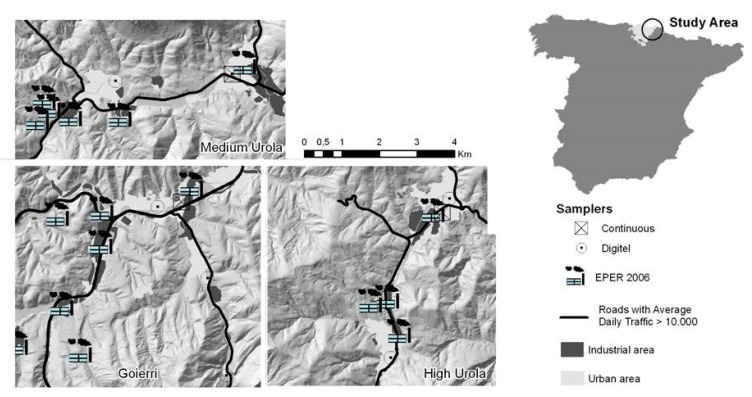
Study area and location of the particles with a diameter < 2.5 µm (PM_2.5_) air samplers.

**Figure 2 ijerph-16-04381-f002:**
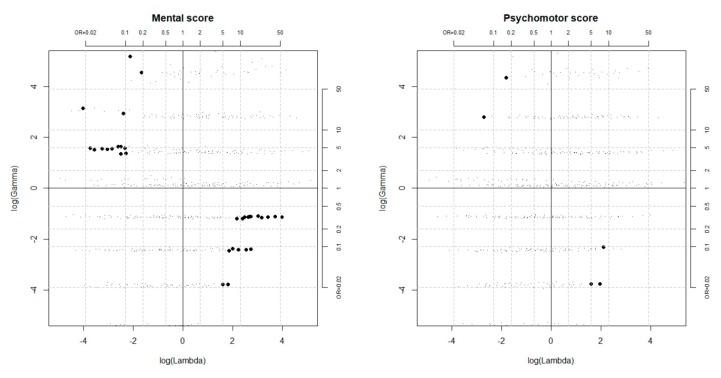
Configurations of the outcome and selection effects (*Γ* and *Λ*) driving the causal effect estimate to zero, as obtained from the simulation-based sensitivity analysis ^a^. Bold points correspond to unobserved confounders that could kill the effect. ^a^ Simulations were performed setting the tuning parameters $p_{10},\;p_{11},\;p_{01},\;p_{00}$ to 0.1, 0.25, 0.5, 0.75, 0.9.

**Table 1 ijerph-16-04381-t001:** Distribution of the characteristics of the child–mother couples in the entire data set and in the subset of the complete observations by exposure.

	N. of Missing Values		Entire Data Set (*n* = 438)		Complete Cases (*n* = 391)	
			Percentage/Mean		Percentage/Mean	
			Treated	Controls	Treated	Controls
Gender	0	Male	45.7	44.7	47.6	45.4
		Female	54.3	55.3	52.4	54.6
Neuropsychologist	0	First	98.4	47.8	97.8	50.6
		Second	1.6	52.2	2.2	49.4
Maternal fruit and veg. intake	5	>405 g/day	76.2	70.6	77.7	70.4
		≤405 g/day	23.8	29.4	22.3	29.6
Maternal smoke	13	Yes	18.2	28.3	18.4	27.4
		No	81.8	71.7	81.6	72.6
Smoke at 32 weeks of pregnancy	34	Yes	11.3	11.0	11.0	11.1
		No	88.7	89.0	89.0	88.9
Nursery attendance	26	Yes	48.6	44.6	48.7	44.9
		No	51.4	55.4	51.3	55.1
Caregiver	28	Mother	55.6	48.2	55.2	48.0
		Other	44.4	51.8	44.8	52.0
Maternal education	2	University	47.6	53.9	44.8	53.0
		Secondary school or less	52.4	46.1	55.2	47.0
Maternal work	0	Manual	45.6	38.3	46.0	36.6
		Non manual	54.4	61.7	54.0	63.4
Mother’s age	0	(mean)	31.2	31.7	31.3	31.7
Mother’s Body Mass Index	1	(mean)	23.1	22.7	23.2	22.8
Parity	0	1+	46.4	44.4	45.6	42.8
		0	53.6	55.6	54.4	57.2
Breastfeeding	24	No	46.6	47.6	47.1	45.5
		Yes	53.4	52.4	52.9	54.5

The reported proportions (for the categorical variables) and means (for the continuous variables) come from the combination of the same quantities arising from the 5 imputed data sets [32].

**Table 2 ijerph-16-04381-t002:** Results of the simulation-based sensitivity analysis for mental score.

Observed Confounder Mimicked by U	ATT	90% CI	Outcome Effect (*Γ*)	Selection Effect (*Λ*)
Gender: male, female (ref)	−1.88	−7.12; 3.35	0.57	1.07
Neuropsychologist: first, second (ref)	0.78	−7.02; 8.57	0.20	>10^3^
Maternal fruit and vegetable intake: ≤405 g/day (ref), >405 g/day	−1.89	−7.24; 3.47	1.04	1.55
Smoke: yes, no (ref)	−1.73	−7.11; 3.65	1.18	0.62
Smoke at 32 weeks of pregnancy: yes, no (ref)	−2.07	−7.35; 3.22	1.76	1.19
Nursery attendance: yes (ref), no	−1.88	−7.13; 3.37	1.05	0.89
Caregiver: both parents, other (ref)	−1.81	−7.16; 3.55	1.75	0.90
Caregiver: mother, other (ref)	−1.85	−7.14; 3.44	0.85	1.42
Caregiver: relative, other (ref)	−2.11	−7.28; 3.07	0.39	0.74
Maternal education: secondary school or less, university (ref)	−1.74	−7.13; 3.65	0.74	1.43
Maternal work: non-manual worker (ref), manual worker	−1.75	−7.12; 3.61	0.63	1.52
Mother’s age: <25, 25+ (ref)	−2.12	−7.29; 3.06	0.31	3.00
Mother’s age: <35 (ref), 35+	−1.92	−7.25; 3.41	0.70	0.58
Body Mass Index: Normal weight, other (ref)	−1.89	−7.22; 3.43	2.25	1.00
Parity: 0 (ref), 1+	−1.92	−7.16; 3.32	0.97	1.15
Breastfeeding: no, yes (ref)	−1.88	−7.18; 3.42	1.67	1.16

Results of the simulation-based sensitivity analysis on the average causal effect of treatment on the treated (ATT) for the mental score. Unobserved confounders U were simulated to mimic the observed covariates. The estimated ATTs (90% CI) after adjusting for the different versions of U are reported, as well as the odds ratios for outcome and selection effects (*Γ* and *Λ*) calculated from the simulations. All the reported results arose from the combination of the results from 5 multiple imputed data sets, according to Rubin [32].

**Table 3 ijerph-16-04381-t003:** Results of the simulation-based sensitivity analysis for psychomotor score.

Observed Confounder Mimicked by U	ATT	90% CI	Outcome Effect (*Γ*)	Selection Effect (*Λ*)
Gender: male, female (ref)	−2.59	−8.13; 2.95	0.58	1.12
Neuropsychologist: first, second (ref)	−0.88	−9.38; 7.61	0.31	>10^3^
Maternal fruit and vegetable intake: ≤405 g/day (ref), >405 g/day	−2.70	−8.32; 2.91	1.60	1.65
Smoke: yes, no (ref)	−2.33	−7.97; 3.31	2.42	0.63
Smoke at 32 weeks of pregnancy: yes, no (ref)	−2.89	−8.35; 2.57	3.77	1.27
Nursery attendance: yes (ref), no	−2.69	−8.28; 2.89	1.07	0.90
Caregiver: both parents, other (ref)	−2.67	−8.19; 2.84	1.62	0.90
Caregiver: mother, other (ref)	−2.60	−8.22; 3.02	1.03	1.44
Caregiver: relative, other (ref)	−2.88	−8.31; 2.54	0.36	0.83
Maternal education: secondary school or less, university (ref)	−2.47	−8.19; 3.25	0.93	1.48
Maternal work: non-manual worker (ref), manual worker	−2.44	−8.14; 3.26	0.71	1.53
Mother’s age: <25, 25+ (ref)	−2.91	−8.28; 2.45	4.30	3.18
Mother’s age: <35 (ref), 35+	−2.70	−8.39; 2.99	0.71	0.57
Body Mass Index: Normal weight, other (ref)	−2.77	−8.2; 2.67	0.84	0.95
Parity: 0 (ref), 1+	−2.69	−8.26; 2.89	1.80	1.23
Breastfeeding: no, yes (ref)	−2.66	−8.16; 2.85	0.76	1.07

Results of the simulation-based sensitivity analysis on the average causal effect of treatment on the treated (ATT) for the psychomotor score. Unobserved confounders U were simulated to mimic the observed covariates. The estimated ATTs (90% CI) after adjusting for the different version of U are reported, as well as the odds ratios for outcome and selection effects (*Γ* and *Λ*), calculated from the simulations. All the reported results arose from the combination of the results from 5 multiple imputed data sets, according to Rubin [32].

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
