# Peer review of "Causal Effects of Prenatal Exposure to PM2.5 on Child Development and the Role of Unobserved Confounding"

_ijerph, 2019, doi:10.3390/ijerph16224381_

Round 1
Reviewer 1 Report
In fact, this manuscript be a good contribution because the nature of the problems related to confounder variables is not fully explained principally when full weight on variables observed are assumed, however, although those observed variables analysis is rationed to factors when may have information in this regard, there is always quantitative uncertainty about their influence on the problem. By it mentioned in introduction section address limitation of study and justify why the presence of unobserved confounding parameters should be used on this epidemiological studies similar than treated here. Please taking in to account that in different sections of manuscript, there are multiple words linked or united, review the wording. Is important that authors mention if the observed risk is due to the continuous presence of raised centering to study particles of diameter <2.5 μm (PM2.5) of airborne, this in order to justify the present work. Please, could consider the expression where is understand that mothers were manipulated by the authors to be espoused, and I think it was not so. Since it transgresses the individual guarantees of each person, the paragraph have to written in a different way, whit aim to not hurt susceptibility over monitoring forms, where anyone it have to be done with discretion and respect by human health.
To authors
This observational study, and I consider is important because contrite to understand an equilibrium into observed and not observed variables trending used combined for solve a dynamic of problem which can help to previse risk under a possible damage to human health. But there are some situations that have to be taken into a consideration if wanted eradicate an uncertainly phenomena that could be introduce by many causes a bias on results and reflected when comparison between groups is done.
However and indistinctly, that maybe the study is other of first works where confounding effect are applied for describe an epidemiological problem related in exposing human to risk, there are many consideration that have to be taken into account writes with details as commentaries inside of manuscript.
Apparently the variables are not random and this situation maybe will reflect many variations between the results abstained on each group. In order to avoid a not synchrony in results is necessary to found a response concordant inter variables group to group. This must be justified with the aim to validate the result because these imbalances in the covariates will be introduce bias on estimate variables under the problem of use prospecting score method, where exist a probability to being able to necessarily assign to observed parameters characteristics of control group.
By other hand, is much complex to accelerate an acceptance on results and attribute to confounding variables much range of solution as here is presented within a problem to be solved, and hence the limitation of the models made with the inclusion of unobserved variables since they tend to be hypothetical.
So I feel that should clearly justify and to mentioning use of this type of variables in epidemiological studies, because as it says with certainty; there are few works related to this methodology that address issues within epidemiology risk.

Author Response
L43: Please mention if the observed risk is due to the continuous presence of dust that is raised by numerous vehicle traffic through roads. In other hand.. Why exclusively the study is centering to study particles of diameter <2.5 μm (PM2.5) of airborne, this in order to justify the present work.
The province of Gipuzkoa is characterized by the presence of iron and steel industries and unexpected high levels of PM2.5, considering the small number of inhabitants of the municipalities in the area (maximum 13.000 inhabitants). Additionally, the concentrations of neurotoxic elements
in the particulate matter is higher than in the European cities (Lertxundi et al, 2010). The NO2 levels, on the contrary, are very low. It is for this reason that in this analysis we focused on PM2.5.
L73: Note: Taking in to account that in differents sections of manuscript, there are multiple words linked or united, is important to review the wording .
Thank you for noticing this point. We have revised the manuscript and everything now should be in place.
L80: Establish a justification why Bayley's scale of child development is used to confine this study only to children who have fifteen months old.
In this paper, we used tests and scales defined in the INMA project (Children and the Environment) and common to all the enrolled cohorts. In particular, as in Lertxundi et al. (2015), we considered children scores at the first follow-up. We refer to the INMA protocol for details, adding the following reference:
Ramón R, Ballester F, Rebagliato M, Ribas N, Torrent M, Fernández M, Sala M, Tardón A, Marco A, Posada M, Grimalt J, Sunyer J; Red INMA. The Environment and Childhood Research Network ("INMA" network): study protocol. Rev Esp Salud Publica. 2005 Mar-Apr;79(2):203-20. [Article in Spanish]
L84: Which was the consideration for applying the questionnaire only in the first and third trimesters of pregnancy in Mothers.
In the INMA study it was decided to perform 2 questionnaires, one at the beginning and one at the end of pregnancy, in order to collect longitudinal information on different groups of variables of interest. Two time points were sufficient to capture possible changes in those characteristics susceptible to change. We refer to the INMA protocol for details (see our response to the previous question).
L85: Take into account that in the random assignment of treatments where a division of individuals is done, it action must be balanced in terms of the base covariates, if they are not alineated, will be biases in results. So in this section or in methodology part must be done a mention of this relationship.
In the methods section we write: “Once defined the relevant covariates for the PS estimation, the key criterion driving the specification of the PS model was to obtain PS estimates that balanced the covariates distribution between treated and controls [9]. According to the procedure implemented in the STATA command pscore, we checked the balancing property of the PS by comparing the covariate distribution between the two groups of exposure within strata of units with similar estimated PS.”

Reviewer 2 Report
This paper is essentially a statistical modelling investigation into whether plausible ‘virtual’ confounders used on previous collected epidemiological data used in an investigation of strength of association between maternal exposure to PM2.5 and subsequent toddler mental acuity could be sufficient to remove the association between exposure and outcome. A number of standard confounding variables were already included in the previous epidemiological study.
It should be noted that the confidence intervals for the associations are in fact not significant in the original epidemiological model: even the 90% confidence interval range quoted encompasses the zero association. However, that fact aside, the authors provide a clear description of the methodology they followed that allowed them to conclude that their association estimates are robust to the presence of unmeasured confounders having strength similar to that of one of the observed covariates.
If the authors can justify why there is merit in pursuing their goal of testing robustness of association even though the association is non-significant then I would be happy to support publication in IJERPH, which is an appropriate journal for this sort of work.
Other than the above comment, this paper is fluently written and clearly presented. I only draw attention to the following very few typos.
L28: insert “of” to read “to that of one…”
L63: delete “of” to read “despite the importance of…”
L77: delete “of” to read “was approximately…”
L83: the word “features” reads oddly here; how about “characteristics”?
L226: rewrite middle part of sentence to read “indicating that it was possible to find, for each treated case, a matched control with…”
L257: insert space before “when”
L311: delete “of” to read “despite the large…”
L337: delete “to” to read “could have brought a certain…”
L354: delete “be” to read “could use…”
L356: replace “allow” with “permits”
Author contributions: What did author Jesus Ibarluzea do? This person is not mentioned at all in the author contribution section.
Did the study have ethics clearance, and assuming so should there be a statement about this in this paper?
Author Response
We thank the reviewer for the comments.
Point-by-point response:
If the authors can justify why there is merit in pursuing their goal of testing robustness of association even though the association is non-significant then I would be happy to support publication in IJERPH, which is an appropriate journal for this sort of work.
Thank you for your comment which concerns a very important point. In order to make clearer our position about the opportunity to perform this kind of sensitivity analysis also in case of “non significant” associations, we modified the original paragraph in the Discussion section which addressed this issue:
“In interpreting the results of the sensitivity analysis, we focused on the point estimates of the effects and not on the confidence intervals limits as sometimes suggested (that means searching for possible unobserved confounders able to explain away the upper or lower limit of the interval, depending on the sign of effect estimate). This choice was motivated by our will to distinguish between evaluation of bias and evaluation of uncertainty. In particular, we think that the sensitivity analysis should be performed regardless the wideness of the confidence interval and its position in respect to the null hypothesis, because its aim is to evaluate the possible presence of major bias due to omission of relevant confounders, not the “significance” of the results. When the confidence interval of the estimate is wide as in our analysis, the robustness of the point estimate, if demonstrated, can confer a certain strength to the conclusions, despite of the large sampling variability. As other kinds of selective reporting, limiting the sensitivity analysis to the subset of “significant” results may produce a bias in the literature, with only studies with narrow confidence intervals evaluated for their robustness in respect to the unobserved confounding. This issue is crucial especially in a perspective where, as desirable, this kind of procedures can become standard in observational studies.”
Author contributions: What did author Jesus Ibarluzea do? This person is not mentioned at all in the author contribution section.
Jesus Ibarluzea is the principal investigator of the INMA study, he contributed in reviewing the article and in discussing the results. We now explicitly specify his contribution in the Contribution section.
Did the study have ethics clearance, and assuming so should there be a statement about this in this paper?
We added the following sentence in the paper (lines 102-103): “The study was approved by the hospital ethics committee and all participating mothers provided informed consent”.
Reviewer 3 Report
This manuscript is essentially a statistical evaluation of the potential impact of unmeasured confounding on the widely observed association between maternal exposure to PM2.5 during pregnancy and infant neuropsychological development. Data were obtained from a subset of the 391 children in a previously published study conducted in the Basque region of Spain. Propensity matching was used to control for a large set of potential confounders, and the impact of an unmeasured confounder was estimated at varying levels using simulations. The results suggest that the results were robust at most levels of the unmeasured confounder, and only a very strong unmeasured confounder would appreciably alter the relationship between PM2.5 and infant neuropsychological status.
The manuscript is generally well written, albeit highly technical for non-statistical readers. I am an epidemiologist, not a biostatistician, but I am familiar with the methods used (MI, simulations) and believe that they are appropriately applied. The findings are useful in that they provide quantitative proof of the generally held assumption that moderate to strong associations between a given exposure and health outcome are unlikely to be the result of unmeasured confounding unless the confounding is very strong. As noted below, I have several concerns that need to be better addressed, most importantly the lack of precision due to high variability
Specific Comments
1. The authors should note that their approach to causal analysis is similar to the potential outcomes (or counterfactual) approach which specifies in detail what would happen under hypothesized alternative patterns of interventions or exposures (Hernan, 2005).
2. The use of MI to estimate the 30% missing weekly PM2.5 averages seems reasonable, but why also impute five values of the average level of PM2.5 during the entire pregnancy (lines 96 – 98) instead of simply using the actual monitoring data in combination with the imputed 30% missing values?
3. Many of the covariates were categorized into crude binary categories (lines 145 – 151), e.g., secondary school or less versus university; non-manual worker versus manual worker. This simplified the propensity scoring but may result in pairs that are only superficially similar. It may also explain the overlap between the treatment and control groups (lines 225 – 227).
4. Although the point estimates for PM2.5 regarding both the mental and psychomotor scores are negative, the confidence limits are very wide and include the null value of zero, even when set at the 90% instead of the more usual 95% level. The authors note that this is due to the large within and between-imputation variances (lines 242 – 243), but such variability raises concerns about the precision of their approach. More information is needed about the reasons for such variability and how it can be reduced for the approach to be widely applied in epidemiological studies.
5. Γ and Λ apparently are odds ratio with the outcome variable Y set the 75th percentile or zero (lines 248 – 249) but it is not clear how the relate to “outcome and selection effects” as noted in Table 1 and 2.
Author Response
We thank the reviewer for the comments.
Point-by-point response
The authors should note that their approach to causal analysis is similar to the potential outcomes (or counterfactual) approach which specifies in detail what would happen under hypothesized alternative patterns of interventions or exposures (Hernan, 2005).
As argued by the reviewer, our approach follows the potential outcome approach. In fact, in the Notation section of the paper, we write: “According to the potential outcome approach to causal inference, under the Stable Unit Treatment Value Assumption, we associated to each child two potential outcomes Yi(1) and Yi(0), representing the score under treatment and under control. For each child it was possible to observe only one of the two potential outcomes: for children belonging to the treated group we observed only Y(1) (Yi(1)=Yi if Ei=1); for children belonging to the control group we observed only Y(0) (Yi(0)=Yi if Ei=0)”, citing the book by Imbens and Rubin (2015).
The use of MI to estimate the 30% missing weekly PM2.5 averages seems reasonable, but why also impute five values of the average level of PM2.5 during the entire pregnancy (lines 96 – 98) instead of simply using the actual monitoring data in combination with the imputed 30% missing values?
This is what we did. On the other hand, it should be considered that in this specific application, the time scale on which we performed multiple imputation (weekly data) is different from the time scale of the exposure (pregnancy time). This is the reason why the procedure resulted in 5 multiple imputed values of exposure for each mother/child. The idea underlying multiple imputation is generating several (5 in our article) different versions of the data set, where the missing entries are replaced by values sampled from appropriate distributions Therefore, we expect that, if the information to calculate to average exposure during pregnancy is in part missing (in our case weekly data were characterized by 30% of missing values), we will obtain, at the end, 5 different imputed versions of the exposure for that woman/child. In the new version of the paper, we made small changes to the Data section in order to better explain the procedure, which is however described in more detail in Lertxundi et al. (2015), which we referred to.
“Due to the presence of moving monitors, daily measurements were characterized by the presence of by design-missing values. With the aim to assess the average exposure of the mother-child couples during pregnancy, we used a Multiple Imputation procedure starting from the weekly data. First, we derived weekly means from the daily levels of PM2.5 measured in each location. Considering a weekly mean as missing when daily measurements were missing for more than two days during the week, the percentage of missing weekly averages during the study period was around 30%. Then, we applied a Multiple Imputation (MI) procedure [21], under the Missing At Random (MAR) assumption, on the incomplete data set including weekly averages concentrations of the air pollutants measured by the monitors, meteorological variables and seasonality indicators. In this way, we generated 5 imputed data sets, thus 5 imputed values for each missing weekly mean. Finally, we averaged over weekly means, obtaining 5 imputed values of the average level of PM2.5 during the pregnancy for each mother-child couple. We assumed that the mother-child couples were exposed to the PM2.5 levels measured at the monitoring site located in their town or in the closest one.”
Many of the covariates were categorized into crude binary categories (lines 145 – 151), e.g., secondary school or less versus university; non-manual worker versus manual worker. This simplified the propensity scoring but may result in pairs that are only superficially similar. It may also explain the overlap between the treatment and control groups (lines 225 – 227).
We understand the point raised by the reviewer. Using binary versions of categorical variables with more than 2 levels avoids problems related to sparsity (sometimes the number of subjects in each single category was quite small) at the price of reducing similarity of the matched units. We think that our model is a good conpromise which allows finding a matched control for each treated child, accounting for relevant confounders such as mother’s education and job type. And the results of the sensitivity analysis seem to confirm the there is not major residual confounding. Moreover, we would like also to stress that the model for propensity score described in the paper results from a selection procedure over alternative linear predictors, which eventually included also different specification of the categorical variables (the selection was aimed to find a model satisfying the balancing property of the PS, as described in the Methods section).
Although the point estimates for PM2.5 regarding both the mental and psychomotor scores are negative, the confidence limits are very wide and include the null value of zero, even when set at the 90% instead of the more usual 95% level. The authors note that this is due to the large within and between-imputation variances (lines 242 – 243), but such variability raises concerns about the precision of their approach. More information is needed about the reasons for such variability and how it can be reduced for the approach to be widely applied in epidemiological studies.
Propensity score matching after having defined a binary version of the exposure is a simple way to reply to the research question: “what would have been the mental and psychomotor development of the children with prenatal exposure µg/m3, if their exposure were been µg/m3?” and allowed us to use the Ichino’s procedure for sensitivity analysis as implemented in the sensatt function of STATA. Moreover, this approach has the advantage of not relying on assumptions about the exposure-response relationship, but at the same time, its “non parametric” nature may lead to poor efficiency, thus larger confidence intervals in respect to, for example, a regression model. To better clarify this point, we made few changes in the Discussion section, where we compared our results with the results in Lertxundi et al. (2015): “In Lertxundi et al. [8], the linearity assumption on the exposure-outcome relationship, together with the parametric specification of the exposure-confounders-outcome relationship, contributed to increase the precision of the effect estimates, but, at the same time, could have generated a certain amount of bias due to problems of inappropriate model specification. The PS matching protected us from possible bias due to nonlinearity or inappropriate modelling of the exposure-confounders-outcome relationship, even if at the price of lower estimates precision. It protected us also from bad extrapolation, through a careful check of the substantial overlap of the background variables distributions in the treatment and control groups. Moreover, checking the balancing property for the estimated PS guided us to the best adjustment and helped us in detecting possible covariates to pay special attention to.”
Γ and Λ apparently are odds ratio with the outcome variable Y set the 75thpercentile or zero (lines 248 – 249) but it is not clear how the relate to “outcome and selection effects” as noted in Table 1 and 2.
Yes, they are. In the new version of the paper, we wrote the Γ and Λ are odds ratios and changed the column headers and the note in Table 1 and 2 to clarify the meaning of Γ and Λ.
Reviewer 4 Report
The paper entitled “Causal effects of prenatal exposure to PM2.5 on child 3 development and the role of unobserved 4 confounding: analysis on a birth cohort from the 5 province of Gipuzkoa”
The authors have indicated that prenatal exposure 359 to high levels of PM2.5 had a negative effect on child’s mental and psychomotor development in the 360 analysed cohort, but there is no novelty here and I am not satisfied with their findings and discussion. There are many studies which have showed that high levels of Pm2.5 have negative impact on human health, particularly on children.
I think the authors should think more and add more analysis and results to the paper.
In addition, there some more comments on the manuscript.
Title: it’s too long. It should be modified and tried to make it shorter with delivering the message.
Abstract:
Please write clear aim of the study .
Add KEY CONCLUSION FROM EACH OF SECTIONs of the results and discussion.
Introduction:
Authors haven’t stated good literature study. I would suggest to cite more relevant studies and make a table for all of them.
Line 66-72: the aim of the study isn’t clear and need to be re-written the paragraph.
Methods:
Please add a map of the city
Please move some materials to the supplementary materials as they are too much in the present form.
Result:
Very weak statement without supporting them by previous studies. Please make sure cite the relevant studies and compare with your results and findings.
In the current form, it’s not interested to reader.
It would be great if authors re-write this section and show them in scientific way.
Discussion:
Please add more Figure to support your discussions. I couldn’t see any interesting figure in the manuscript. The Authors started that there some Figures in the supplementary materials, but I could find them. It would be good to bring all figures and tables to the manuscript as there is few figures. So please show Table A1, A2, and A3, Figure A1, A2 and A3 and other to the manuscript.
Conclusions:
Please re-write and change the structure to bulling points. You can write brief conclusion for each section of the results and discussion. Please add the gap study and remain research question in this filed.
Author Response
We thank the reviewer for the comments.
Point-by-point response
I think the authors should think more and add more analysis and results to the paper.
We included Figure A1 and Table A1 in the paper.
Title: it’s too long. It should be modified and tried to make it shorter with delivering the message.
We made the title shorter: “Causal effects of prenatal exposure to PM2.5 on child development and the role of unobserved confounding”.
Abstract:
Please write clear aim of the study. Add key conclusion from each of sections of the results and discussion.
Thank you for the comment. We modified the abstract in order to make our aim clearer and report key conclusions from the Results and Discussion sections:
“Prenatal exposure to airborne particles is a potential risk factor for infant neuropsychological development. This issue is usually explored by regression analysis, under the implicit assumption the all relevant confounders are accounted for. Our aim is to estimate the causal effect of prenatal exposure to high concentrations of airborne particles with diameter < 2.5 µm (PM2.5) on children’s psychomotor and mental scores in a birth cohort from Gipuzkoa (Spain), and investigate the robustness of the results to possible unobserved confounding. We adopted propensity score matching approach and performed sensitivity analyses comparing the actual effect estimates with those obtained after adjusting for unobserved confounders simulated to have different strength. On average, mental and psychomotor scores decreased of -2.47 (90% CI: -7.22; 2.28) and -3.18 (90% CI: -7.61; 1.25) points when the prenatal exposure was ≥ 17 μg/m3 (median). These estimates were robust to the presence of unmeasured confounders having strength similar to the observed ones. The strength required for an unobserved confounder to drive the estimates to zero was very high, thus implausible. The sensitivity analyses results conferred solidity to our findings, despite the large sampling variability. This kind of sensitivity analysis should be routinely implemented in observational studies, especially in exploring new relationships.”
Introduction:
Authors haven’t stated good literature study. I would suggest to cite more relevant studies and make a table for all of them.
According to the reviewer comment, we updated the cited literature. Being our article a research article and not a review, we think that citing and describing the literature in the Introduction section is sufficient.
Line 66-72: the aim of the study isn’t clear and need to be re-written the paragraph.
We tried to improve the aim description:
“In the present paper, with reference a the birth cohort of the province of Gipuzkoa [8], after having estimated the causal effect of high PM2.5 exposures during maternal pregnancy on child mental and psychomotor outcomes at 15 months of age, we checked the robustness of the results to the possible presence of unobserved confounders through a sensitivity analysis based on simulations. In particular, we compared the actual causal effect estimates obtained by using propensity score (PS) matching [9], to those obtained after controlling also for simulated unobserved confounders having different strength, according to the method proposed by Ichino et al. [11] and implemented in the sensatt function of STATA [19,20].”
Methods:
Please add a map of the city
We are analysing a province including several municipalities. The map of the area has been moved from the Supplemental material to the main manuscript.
Please move some materials to the supplementary materials as they are too much in the present form.
We understand the reviewer feeling, but we think that moving part of the Material and Methods section to the supplementary material could make difficult the comprehension of the sensitivity analysis rational to an interested reader.
Result:
Very weak statement without supporting them by previous studies. Please make sure cite the relevant studies and compare with your results and findings. In the current form, it’s not interested to reader. It would be great if authors re-write this section and show them in scientific way.
We reported in the Results section the results of the main analyses and of the sensitivity analyses in a scientific way. Comparison with the literature, in particular with Lertxundi et al. (2015), can be found in the Discussion section. Regarding the alleged weakness of our statement, in our opinion - on the contrary - our results have two strengths: 1) we used an approach different from the usual one (regression analysis) to estimate the casual effect of prenatal exposure to PM2.5 on child’s development, and our results, despite of the large confidence intervals, indicate the existence of an effect. This strengthens the hypothesis, supported by the existing literature, that there is an effect, because the results are not dependent from the statistical method; 2) our sensitivity analysis found that the results were robust to the presence of unobserved confounders. This means that the set of confounders selected is appropriate to remove confounding in this specific analysis, but possibly also in other case studies.
We added the following sentences in the Discussion section:
“…even if it refers to a specific case study, the sensitivity analysis may provide insights about the confounders to be accounted for in future studies on the same research question.“
“Anyway, the fact that different statistical methods brought to consistent results strengthens our conclusions.”
Finally, we think that the strength of the results should not be a criterion to decide about the opportunity to report them. Selective results reporting and publishing (for example by focusing only on those ones which are supported by previous studies) is a bad practice that may generate publication bias.
Discussion:
Please add more Figure to support your discussions. I couldn’t see any interesting figure in the manuscript. The Authors started that there some Figures in the supplementary materials, but I could find them. It would be good to bring all figures and tables to the manuscript as there is few figures. So please show Table A1, A2, and A3, Figure A1, A2 and A3 and other to the manuscript.
After your suggestions we moved Figure A1 to the paper.
Conclusions:
Please re-write and change the structure to bulling points. You can write brief conclusion for each section of the results and discussion. Please add the gap study and remain research question in this filed.
Thank you for the suggestion that we considered. However, after checking the latest published papers in IJERPH, we found that the Conclusions are not usually organized in bulling points. Thus, we decided to leave the section in the original form.
Round 2
Reviewer 1 Report
Dear Authors,
I already the manuscript was reviewed, Respect to previously observations proposed, all observed were applied. On a corroboration with both documents , I observed that not only applied my recommendations, also adhered more discussion according to other recommendation done by the others reviewers selected by this journal, and so with this extended now have a particular discussion on all parts of document and is more understandable than the first revision.
Regards
Author Response
Thank you for your useful comments.
Reviewer 3 Report
In general, the authors have been responsive to my concerns. A few a additional comments:
The fact that the approach is similar to the potential outcomes (or counterfactual) approach needs to be stated in the Introduction, not buried in the Methods section under Notation, and its relevance to causal analysis in epidemiological studies as noted by Hernan, 2005 must be cited. I remain concerned about the lack of precision in the effect estimates. The authors reply in very statistical terms about how their approach helps to minimize problems due non-linearity and inappropriate modeling of the of the exposure-confounder-outcome relationship, but does not address the question of what can be done to produce more precise estimates. To be most useful to epidemiologists and policy makers, a method needs to produce estimates that are both unbiased and precise. Although the authors have noted that Γ and Λ are odds ratios, it is still not clear why they represent "outcome" and "selection" effects. Again, a more non-statistical explanation of what they mean is needed.
Author Response
1) The fact that the approach is similar to the potential outcomes (or counterfactual) approach needs to be stated in the Introduction, not buried in the Methods section under Notation, and its relevance to causal analysis in epidemiological studies as noted by Hernan, 2005 must be cited.
In the new version of the paper we cite Hernan 2005 in the introduction section and we explicitly write that we use a potential outcomes approach to casual inference.
2) I remain concerned about the lack of precision in the effect estimates. The authors reply in very statistical terms about how their approach helps to minimize problems due non-linearity and inappropriate modeling of the of the exposure-confounder-outcome relationship, but does not address the question of what can be done to produce more precise estimates. To be most useful to epidemiologists and policy makers, a method needs to produce estimates that are both unbiased and precise.
In our opinion, the way we can obtain more precise estimates is by using a generalized propensity score (GPS) approach. We modified the last sentence in the paper to connect the use of GPS (in future promising analyses) with the issue related to sampling variability.
3) Although the authors have noted that Γ and Λ are odds ratios, it is still not clear why they represent "outcome" and "selection" effects. Again, a more non-statistical explanation of what they mean is needed.
In the new version of the paper we explain better the meaning of the two odds ratios.
Reviewer 4 Report
Thanks The Authors have address all my comments. well done.
Author Response
Thank you for your comments.